# Knowledge and Perceptions of Healthcare Workers about the Implementation of the Universal Test and Treat Guideline in Under-Resourced, High-HIV Prevalence Rural Settings

**DOI:** 10.3390/healthcare11070968

**Published:** 2023-03-28

**Authors:** Lerato Martina Maluleka, Naomi Hlongwane, Mathildah Mpata Mokgatle

**Affiliations:** Department of Epidemiology and Biostatistics, School of Public Health, Sefako Makgatho Health Sciences University, Pretoria 0208, South Africa

**Keywords:** HIV, Universal Test and Treat, primary care, South Africa

## Abstract

Background: South Africa (SA) began implementing its Universal Test and Treat (UTT) policy in September 2016 and Same Day Initiation (SDI) in 2017, aiming to meet the UNAIDS 90-90-90 targets by 2020. With significant advances in HIV testing, large gaps remain in the linkage and retention in care. As part of a contribution to the successful implementation of UTT, this study aims to examine progress in the implementation of the UTT and to identify gaps and facilitators in the successful implementation of the guidelines from the perspective of healthcare providers from under-resourced, high-HIV prevalence rural settings. Methods: We conducted a census of all 170 professional nurses from 18 primary healthcare (PHC) clinics in Rustenburg, South Africa, between October 2018 and February 2019. The perceptions, knowledge and attitudes of nursing staff associated with UTT implementation were investigated though the dissemination of self-administered questionnaires. Stata 16.0 was used to analyse the data. Frequency and contingency tables were used to present categorical data. The precision of the estimates was measured using a 95% confidence interval (95% CI), and the *p*-value of statistical significance is *p <* 0.05. Results: The facilities were found to have adequate governance and supervision, but gaps were identified, including staffing challenges, bottlenecks and under-resourced service delivery platforms. It was found that a high level of knowledge is a predictor of positive perception of the UTT programme and its implementation. Being supported by capacity development and having positive perceptions of UTT were important motivators for UTT implementation. Conclusions: This study was able to identify potential facilitators of the UTT strategy implementation at the selected facilities. Clinical guidelines and policies on UTT contributed to successful implementation, which means that the process of closing the gaps identified should prioritise the delivery, support and prioritisation of capacity development, infrastructure and the provision of clinical guidelines to all healthcare workers. It is recommended that nurses receive training on UTT and its benefits to increase their knowledge and promote its successful implementation in clinics.

## 1. Introduction

Eastern and Southern Africa remain the regions hardest hit by HIV, accounting for over half of the global HIV burden [1,2]. Notwithstanding this, notable progress has been made against HIV with the expansion of antiretroviral therapy (ART) and the successive adoption of the Universal Test and Treat (UTT) guideline recommended by the World Health Organization (WHO) [3,4,5]. Conversely, health systems in sub-Saharan Africa remain vulnerable and over-burdened [6,7,8]. In addition, persistent deficiencies in health systems have made it challenging for many countries in the region to meet UNAIDS 90-90-90 HIV targets and to take full advantage of UTT policies [8,9,10]. South Africa has the region’s highest HIV burden, with nearly 8 million people living with HIV, of whom 5.5 million are on ART and 5 million are virally suppressed [4]. Despite significant efforts to improve access to treatment, challenges remain in reaching patients and retaining them in care [11,12,13].

South Africa introduced the UTT strategy in 2016 and the Same Day Initiation (SDI) ART policy in 2017, leading to increased testing efforts, but ART initiation within the expected timeframe remains a problem [10,14,15,16]. Healthcare providers play a critical role in the implementation of UTT. Unfortunately, the UTT implementation was not accompanied by expanded human or infrastructural capacity, and effective implementation of the ART expansion guidelines requires a well-trained and adequately staffed workforce [6,7,8,17]. Notwithstanding the efforts to expand HIV testing, South Africa needs an additional three million people to start ART to reach 95% of the people diagnosed with HIV on ART by 2030 [4,18]. As a result of the increased demand for ART, the strain on already overburdened public services may increase, negatively impacting the quality of care. [19,20]. As a result, it is critical to comprehend how public sector healthcare providers manage their implementation of the UTT guidelines, identify gaps and devise solutions to maximise and sustain benefits. The purpose of this study was to look into the perceptions, knowledge and factors associated with the implementation of UTT in underserved rural areas.

## 2. Materials and Methods

### 2.1. Study Design and Population

The study setting for this descriptive cross-sectional design was primary healthcare clinics (PHC) in Rustenburg, South Africa. There were 18 PHC with 8 medical practitioners and 170 professional nurses in total. The study population consisted of professional nurses and healthcare managers employed at the time of data collection, which was from October 2018 to February 2019. With a population size of n = 170, due to response rate issues and the limited size of the research population, a census of all 170 professional nurses was chosen as the sampling technique.

### 2.2. Measures

The data were collected through a structured, self-administered questionnaire. We collected information on the respondents’ socio-demographics, knowledge about UTT and perceptions about UTT. Their knowledge of UTT was assessed through asking 19 questions, which were posed as multiple-choice questions to allow the respondents to select the correct response. Other knowledge questions required the respondent to answer “Yes” or “No”. The correct response was coded “1” and the incorrect or a non-response was coded “0”. A total knowledge score was computed by summing the scores of each question. For each participant, the possible total knowledge score ranged from 0 to 19, including questions on who is eligible for UTT, the period of initiation, the benefits of UTT and what is necessary for the successful implementation of UTT. A three-item Likert scale was employed to quantify their perceptions of UTT. The participants were asked if they would suggest or consider providing UTT, the availability of the resource and how it is being implemented. Responses were categorised as “Yes”, “No” and “Not sure”.

### 2.3. Data Collection

The data were collected using a self-administered, structured questionnaire formulated in English. The questionnaires were distributed to the respondents by trained research assistants. The nurses completed the questionnaires individually but the research assistants were available to check the completeness of the questionnaires. Due to the nature of this new, progressive HIV treatment and care intervention and its novelty, the questionnaire was developed by referring to previous qualitative tools and constructs obtained from a review of the literature on PMTCT and perceptions and implementation of the universal HIV treatment guidelines [21,22]. To ensure validity, the tool was pre-tested to ensure that the respondents understood the questions. This also allowed for the identification of possible problems with the completion of the questionnaire. The research assistants did not encounter any challenges with the completion of the tool by the six healthcare workers who engaged in the pilot, and no changes were made. The pre-test increased the reliability and validity of the study.

### 2.4. Data Analysis

The analysis of the data was performed using the STATA IC version 16.0 statistical package (STATA Corp, College Station, TX, USA). Descriptive statistics such as frequencies, percentages and proportions were computed to describe the study variables. Socio-demographic data were obtained. To determine the respondents’ knowledge of UTT, the responses to the 19 knowledge questions were tabulated and reported as proportions according to correct and incorrect responses, and the mean and standard deviations (SD) computed from the total knowledge score. The mean was used to categorise the knowledge levels. Knowledge scores above the mean were categorised as good knowledge, while scores below the mean were categorised as poor knowledge. The variable gender (male or female) was the dependant variable for the study, and the variables within the constructs of socio-demographics, sexual behaviour, knowledge of STIs and partner notification were the independent variables. The Pearson chi-square test was used to examine the difference in demographic UTT knowledge and UTT perceptions between the respondents who suggested UTT and those who did not. The association between variables was measured by using bivariate logistic regression. A significance level of 5% was accepted. The strength of association between the dependent and independent variables was described using an odds ratio at a 95% confidence interval (CI). Statistical significance was set at *p* < 0.05 for all variables.

### 2.5. Ethical Considerations

The Research Ethics Committee of Sefako Makgatho Health Sciences University (SMUREC/H/294/2017) reviewed the protocol and gave it ethical clearance. The relevant university authorities granted permission to conduct the study. Informed consent was obtained from all respondents. Participation was voluntary, including the right to withdraw from the study without any preconditions. For anonymity, no identifying information was collected and the data file was password-protected, with access limited to the lead investigator.

## 3. Results

### 3.1. The Socio-Demographic and Other Characteristics of the Respondents

One hundred and fourteen (n = 114) healthcare workers were enrolled into the study, among whom there were 94 (82.6%) females and 20 (17.5%) males. The mean age was 42 years, ranging from 23 to 64 years. The ages of the participants were categorised into four groups. Over a third (33.6%) of them fell in the age group 23–35 years, while those who were above the age of 55 constituted just 17 (15.4%) of the study population. With regard to the marital status of the respondents, the majority of them (n = 61, 53.5%) were married. The vast majority (n = 82, 71.9%) of the participants worked in primary healthcare facilities (PHC), while 32 (28.1%) worked in community health centres (CHCs). Of a sample of n = 114 professional nurses, more than half (n = 71, 62.3%) were professional nurses, followed by NIMART providers (n = 31, 27.2%) and operational managers (n = 12, 10.5%), respectively. With regard to their work experience, more than half (n = 59, 55.1%) had over 10 years of experience as a nurse. Furthermore, 51.7% (n = 59) had received HIV training and over a third (n = 32, 31.7%) had had more than 10 years’ experience working with HIV patients (Table 1).

### 3.2. Access to and Availability of UTT Information

Almost all (n = 111, 97.3%) of the participants knew about UTT. The sources of respondents’ information about UTT came more from other healthcare providers and guidelines (n = 78, 68.4%) and (n = 55, 48.2%), respectively, and respondents were allowed to indicate more than one source. The majority (n = 95, 84.1%) stated that they had access to the 2016 national consolidated clinical guidelines and were either mentored or supported on UTT (n = 87, 76.3%), and most indicated that this support came from their supporting partner (n = 62, 71.3%) (Table 2).

### 3.3. Respondents’ Perceptions of the UTT Programme

The perceptions of the respondents towards the UTT approach were assessed based on eleven questions to which the answers could be yes, no or unsure. The results indicate that most respondents had positive perceptions of the UTT (n = 79, 69.3%) with a score equal to or more than the mean score, while 30.1% had negative perceptions, and they scored less than the mean score. Most (n = 101, 88.6%) had considered providing UTT services. However, only 62.8% (n = 71) had suggested UTT services to their patients in the past year. This apparent inconsistency may be linked to the unavailability of the necessary resources, as almost half of the respondents (n = 55, 48.3%) stated that their sub-district did not have sufficient medical supplies to implement the UTT programme. In addition, over half (n = 64, 56.1%) of the respondents thought that the UTT programme could do harm if not carefully implemented. Thus, 48.3% (n = 55) stated that UTT was not ready to be made widely available (Table 3).

### 3.4. Knowledge Level of Respondents about UTT Strategy

The mean of the total knowledge score about UTT on the 19-item question scale was 16 (SD = 2.9), and the range was 4–19. As such, most (n = 80, 70.2%) of the participants had a high level of knowledge of UTT, above a score of 70.5%, whereas those with insufficient knowledge made up 29.8%, with a score of 42.5%. Although most of the respondents were knowledgeable about UTT, over half of the respondents (n = 54, 46.4%) did not see the need to prioritise it for CD4 below 200 and those at HIV stage 4 (Table 4).

### 3.5. Association between Knowledge and Perceptions

There was a significant association between the level of knowledge and perceptions about UTT (*p* = 0.001). The study found that those participants with a higher level of knowledge had good perceptions about the programme (Table 5).

### 3.6. Predictors of Positive Perceptions towards UTT

In determining the factors associated with suggesting UTT to patients, a logistic regression model was deployed whereby different variables were subjected to the model for bivariate logistic regression analysis, and crude odds ratios (COR) were determined at a 95% confidence interval. The variables that were statistically significant (*p* < 0.05) were further subjected to multivariate logistic analysis, and adjusted odds ratios (AOR) were determined. Being supported and having positive perceptions about UTT was significant in the multivariate analysis. The odds of suggesting UTT were four times more likely for respondents who received support (AOR = 1.46, 95% CI: 1.40–14.19). Furthermore, the odds of suggesting UTT were nine times more likely for those who had positive UTT perceptions. In addition, there was no difference in those who received training, had high knowledge of UTT and had UTT guidelines, and the odds ratio was one (Table 6).

## 4. Discussion

This study contributes to the body of literature on the facilitators of the implementation of the UTT guidelines. This study examined the respondents’ knowledge and perceptions of early ART initiation under the Universal Test and Treat guideline, and highlights the role of healthcare workers in advocating decisions about ART initiation.

The findings show a high level of knowledge of UTT among the respondents, most of whom were nurses implementing the UTT programme. Studies have found that healthcare workers generally have good knowledge about health programmes, including HAART programmes [23,24,25]. Most of the respondents reported that their knowledge was peer-led rather than the outcome of formal training, which replicated the findings of other studies. Moreover, the literature provides evidence of the sources of knowledge that could be instilled through the initiative of “task-shifting” from physicians to nurses during the implementation of ART initiation programmes, nurse capacity building projects and the opportunity of high patient headcounts [25,26,27,28,29,30]. Task-shifting and mentorship are a reality in high-HIV prevalence rural settings in South Africa.

The good knowledge score in our study is most likely due to the strong presence of supporting partners in the clinics, such as civil organisations, as well as training and mentoring programmes. Capacity-building through training and mentoring programmes by health managers are referred to by other researchers, and they are associated with an increase in positive attitudes and successful programme implementation [23,31].

The findings show that good knowledge and positive perceptions of the programme were associated with the availability of and access to clinical guidelines on the implementation of the UTT in the clinics. This is in line with Moran et al. (2022), who note that formal sources of information such as guidelines and official memoranda improve the implementation of the programme [29]. Pascoe et al. (2020) also mention that given sufficient time to develop, the capacity of nurses and work experience facilitate the implementation of health programmes and satisfy end-users’ expectations [32].

The data show a significant association between having access to clinical guidelines and knowledge and positive perceptions about the implementation of the UTT. More than half of the respondents said that their primary source of information on the UTT was clinical guidelines. The results also showed that participants who had guidelines had better perceptions than those who did not have any guidelines. Lester et al. (2010) found that poor knowledge of the guidelines resulted in an overall low level of implementation [33].

Perceived barriers against the successful implementation of the UTT in this study included concerns about patient adherence, cost-effectiveness, limitations in the stock of ART and limitations in the number of available staff, limitations that have also been reported in other studies conducted over time [29,34]. Despite the above-mentioned perceived barriers, the respondents in this study had favourable perceptions about UTT and would consider providing it. Most respondents supported the implementation of the programme and believed that it is an appropriate use of resources. Such willingness is necessary to promote the acceptance of the UTT by patients. Positive perceptions and motivation to implement the UTT programme were not common in the past decade, where healthcare providers were reluctant to initiate ART early due to a lack of trust in the sustainability of treatment supplies and increased demands on the already sparse human resources for healthcare.

### Limitations

The cross-sectional design is a limitation in our study. The relatively small number of healthcare workers and clinics included means that the results should be used with caution and the findings should not be generalised to other nurses in other settings. Although a self-administered questionnaire was used and it was expected that the respondents would answer honestly, the fact that the researcher was a known colleague might have influenced them in such a way that they would have given answers which they perceived to be those the researcher would prefer. This study excluded other healthcare professionals involved in the provision of UTT such as doctors and pharmacists since they are available in the clinics only on sessional basis.

## 5. Conclusions

This study has shown overall high levels of knowledge and positive perceptions about UTT among most of the respondents who reported support for the implementation of the programme. Most claimed that they had suggested UTT to patients in the past year. Knowledge of the clinical guidelines and policies on UTT contributed to successful implementation, which implies that the provision of clinical guidelines should be prioritised to all healthcare workers to strengthen implementation. Training on UTT and its benefits is recommended to improve nurses’ knowledge and to promote its successful implementation in the clinics. High levels of knowledge are a predictor of positive perceptions about the UTT programme and its implementation. This means that UTT as a new programme for both nurses and patients has implications for both healthcare provider and patient attitudes, acceptability and willingness, and the consequential retention in the programme. Understanding the relationship between nurses’ capacity, attitude, knowledge of the UTT programme and knowledge of the guidelines for UTT implementation contributes to the body of knowledge about UTT and hopefully to reaching the targets of the UTT programme.

## Figures and Tables

**Table 1 healthcare-11-00968-t001:** Profile of the respondents in the study.

Characteristics		Frequency (%)
Sex	Female	94 (82.5)
	Male	20 (17.5)
Age category Missing n = 4	23–35	37 (33.6)
	36–45	30 (27.3)
	46–55	26 (23.6)
	Above 55	17 (15.4)
Marital status	Married	61 (53.5)
	Single	46 (40.3)
	Widowed	7 (6.1)
Type of facility	CHC	32 (28.1)
	PHC	82 (71.9)
Type of provider	NIMART Nurse	31 (27.2)
	Operational manager	12 (10.5)
	Professional nurse	71 (62.3)
Working experience Missing n = 7	>5 years	35 (32.7)
	6–10 years	13 (12.1)
	Above 10 years	59 (55.1)
Certificate or training in HIV management	No	55 (48.3)
	Yes	59 (51.7)
Years working with HIV patients Missing n = 3	<5 years	45 (44.5)
	6–10 years	24 (23.8)
	>10 years	32 (31.7)

**Table 2 healthcare-11-00968-t002:** Access to and availability of UTT information.

Characteristics		Frequency (%)
Know about UTT	Yes	111 (97.3)
	No	3 (2.6)
Where did you hear about UTT (multiple answers)	Training	36 (31.6)
	Media	27 (23.7)
	Journals	10 (8.8)
	Memorandums	43 (37.7)
	Guidelines	55 (48.2)
	Healthcare provider	78 (68.4)
Access to 2016 national consolidated clinical guidelines at place of workMissing n = 1	Yes	95 (84.1)
	No	18 (15.9)
Supported or mentored on UTT in the past 12 months	Yes	87 (76.3)
	No	27 (23.7)
Who offered support Missing n = 26	Manager	16 (18.4)
	Civil organisation	62 (71.3)
	Both manager and civil organisation	9 (10.3)

**Table 3 healthcare-11-00968-t003:** Respondents’ perceptions about UTT.

Statements	Yes	No	Unsure
You would consider providing UTT services	101 (88.6)	9 (7.9)	4 (3.5)
Have suggested UTT services to patients in the past year	71 (62.8)	28 (24.8)	14 (12.4)
You believe UTT is an appropriate use of health resources	83 (72.8)	17 (14.9)	14 (12.3)
Support the implementation of the UTT programme	91 (79.8)	14 (12.3)	9 (7.9)
The sub-district supply chain system can implement the UTT programme easily	49 (42.9)	30 (26.3)	35 (30.7)
The sub-district has sufficient medicinal supplies to implement the UTT programme	36 (31.8)	48 (42.5)	29 (25.6)
UTT introduces many health risks	33 (28.9)	64 (56.1)	17 (14.9)
UTT should not be pursued any further	18 (15.8)	77 (67.5)	19 (16.7)
UTT is not ready to be made widely available	30 (26.3)	55 (48.3)	29 (25.4)
UTT has the potential to do more harm than good if not carefully implemented	64 (56.1)	37 (32.5)	13 (11.4)
We have an ethical obligation to provide UTT because it is an intervention that could increase life expectancy	90 (79.6)	10 (8.9)	13 (11.5)

**Table 4 healthcare-11-00968-t004:** Knowledge of the respondents on UTT.

Statements	Correct Response	Incorrect Response
Recipients of UTT according to the guidelines		
High-risk patients (False)	101 (88.6)	13 (11.4)
Patients with CD4 < 200 cells (False)	81 (71.1)	33 (28.9)
All HIV+ patients (True)	80 (70.2)	34 (29.8)
HIV-positive pregnant women (True)	91 (79.8)	23 (20.2)
Period of initiating people on ART		
CD4 cell count > 500 (False)	100 (87.72)	14 (12.3)
CD4 cell count > 350 (False)	100 (87.7)	14 (12.3)
CD4 cell count > 200 (False)	95 (83.3)	19 (15.8)
Start regardless of CD4 count (True)	111 (97.4)	3 (2.6)
Prioritise CD4 below 200 and HIV stage 4	53 (46.5)	61 (53.5)
Eligible patients to be assessed for readiness and willingness to start ART	93 (82.3)	20 (17.7)
Early initiation of treatment increases the life expectancy of an individual	104 (91.2)	10 (8.8)
UTT will help reach the UNAIDS 90-90-90 targets	108 (94.7)	6 (5.3)
UTT will help eradicate HIV infections	90 (78.9)	24 (21.1)
UTT will help with EMTCT	102 (89.5)	12 (10.5)
What is necessary for the successful implementation of UTT		
A strong HIV testing service	110 (96.5)	4 (3.5)
Tracing and initiating of pre-ART patients	106 (93.0)	8 (7.0)
A strong supply chain	102 (89.5)	12 (10.5)
Training and mentoring of all prescribers	112 (98.3)	2 (1.7)

**Table 5 healthcare-11-00968-t005:** Association between knowledge and perceptions.

Level of Knowledge	Positive Perceptions	Negative Perceptions	*p*-Value
High	64 (81.0)	16 (45.7)	<0.001
Low	15 (19.0)	19 (54.3)	

**Table 6 healthcare-11-00968-t006:** Predictors of positive perceptions towards UTT.

	Suggested UTT (%)	Did Not Suggest UTT n (%)	COR (95% CI)	AOR (95% CI)
HIV training				
Yes	43 (72.9)	16 (27.1)	Ref	ref
No	29 (52.7)	26 (47.3)	2.41 (1.10–5.26) *	1.10 (0.40–3.02)
Being supported				
Yes	63 (72.4)	24 (27.6)	Ref	Ref
No	9 (33.3)	18 (66.7)	5.25 (2.07–13.28) *	4.46 (1.40–14.19) *
Knowledge of UTT				
High	59 (73.5)	21 (26.5)	Ref	Ref
Low	13 (38.4)	21 (61.7)	4.54 (1.93–10.64) *	1.87 (0.63–5.53)
Having UTT guidelines				
Yes	79 (83.2)	16 (16.4)	ref	Ref
No	7 (38.9)	11 (61.1)	7.76 (2.61–23.06) *	1.02 (0.23–4.42)
UTT Perception				
Positive	63 (79.7)	16 (20.3)	ref	ref
Negative	9 (25.7)	26 (74.3)	11.37 (4.46–29.10) *	9.60 (3.36–27.4) *

COR: crude odds ratio, AOR: adjusted odds ratio, * significant at *p* < 0.05.

## Data Availability

A dataset will be submitted upon request by the editor and the data will be sent to the university’s library data curation server for access.

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
