# Peer review of "Knowledge and Perceptions of Healthcare Workers about the Implementation of the Universal Test and Treat Guideline in Under-Resourced, High-HIV Prevalence Rural Settings"

_healthcare, 2023, doi:10.3390/healthcare11070968_

Round 1

Reviewer 1 Report

The present work is an interesting cross-sectional survey on nurses knowledge and perceptions about HIV Universal Test and Treat strategy in South Africa. Given the high-burden setting and the key role, often overlooked, played by nurses and other non-medical HCW on such programs.

I would recommend the paper for publication on Healthcare upon the resolution of some minor issues:

- Could you describe how the validation of the questionnaire was carried out?

- How do you deal with missing values (if any)?  

- Why you didn't use a "Knowledge, Attitudes and Practices" design for the survey?

Author Response

Reviewer 1 comments

Author response

- Could you describe how the validation of the questionnaire was carried out?

Addressed in line 153-164

- How do you deal with missing values (if any)? 

Missing values are captured in the tables and the analysis excluded the values from the computation

Why you didn't use a "Knowledge, Attitudes and Practices" design for the survey

Since UTT was a new program that was a challenge for both service providers and patients, we thought it best to examine progress in the implementation of the UTT and to identify gaps and facilitators in the successful implementation of the guidelines

Reviewer 2 Report

Line 68: primary health care (PHC) clinics in Rustenburg, South Africa. There are eighteen PHC clinics with 8 medical practitioners, and 170 professional nurses in total. How were these PHC selected and are these private or public facilities?

Limitations

Shortcomings of data being collected through a structured self-administered questionnaire 

Author Response

Reviewer 2 comments

Author response

Line 68: primary health care (PHC) clinics in Rustenburg, South Africa. There are eighteen PHC clinics with 8 medical practitioners, and 170 professional nurses in total. How were these PHC selected and are these private or public facilities?

All primary health care facilities are public facilities and as it was mentioned, the limited number of healthcare workers warranted that we include all the facilities and have a census sample of the nurses available at the time of data collection

Limitations

Shortcomings of data being collected through a structured self-administered questionnaire 

It has already been captured in the data collection section the limitation was averted by having research assistants onsite to ensure completeness of the questionnaire

Reviewer 3 Report

The manuscript itself is ok for publication but lacks high novelty. My question is how does this research facilitate UTT implementation in south Africa? 

Author Response

Reviewer 3 comments

Author response

The manuscript itself is ok for publication but lacks high novelty. My question is how does this research facilitate UTT implementation in south Africa?

UTT as a new program for both nurses and patients has implications for both healthcare provider and patient attitudes, acceptability, willingness and consequential retention to the program and adherence. Understanding the capacity, attitude and knowledge of the guidelines for UTT implementation contributes to the body of knowledge in making progress and reaching targets for universal test and treat.

Reviewer 4 Report

The title is too long, should be more concise.

"The purpose of this study was to look into the perceptions, knowledge, and 64 factors associated with the implementation of UTT in underserved rural areas."

What do you use this data for? What is the utility of the study?

The contain of the study has no practical relevance.

Author Response

Reviewer 4 comments

Author response

The title is too long, should be more concise.

"The purpose of this study was to look into the perceptions, knowledge, and 64 factors associated with the implementation of UTT in underserved rural areas."

Title shortened: Knowledge and perceptions of healthcare workers on the implementation of the universal-test-and-treat guideline,  in under resourced, high HIV prevalence rural settings

What do you use this data for? What is the utility of the study?

Any scientist working thing the field of HIV prevention, control, retention care to care and treatment knows the serious challenges faced by implementers and healthcare providers in achieving full target for UTT and this study adds to the body of knowledge in that area. I disagree with the reviewer

The contain of the study has no practical relevance.

Reviewer 5 Report

I read the manuscript with great interest. I liked the presentation style and discussion of the findings. A few things I may have to suggest. The authors must make sure that they proofread the revised version of the manuscript before uploading it to the journal website. I think almost all tables are incorrectly referred to in the main text, which could have been avoided, had the authors done a careful proofreading.

Secondly, the authors also did not follow a single style when it came to referring to previous studies for in-text citations. For example, sometimes they just used the authors-year, as Moran et al (2022) in line 208. On other times, they used both, as "Lester et al, 2010 found that poor knowledge of guidelines resulted in an overall low implementation [29]" in lines 217-218.

Also, I think, some sentences (e.g., "It is recommended that nurses receive training on UTT and its benefits to increase their knowledge and promote successful implementation in clinics." in lines 35-36) need to be rewritten or rephrased for clarity. For example, it was not clear to me from the way the sentence in lines 35-36 was constructed who was making the recommendation and/or on what basis. If the recommendation is based on the findings of the present study, then it should be made plain and clear. I am not going to list such sentences. However, I encourage the authors to look for sentences that have confusing construct and try to rewrite/rephrase them for clarity. Again, these things could be found in a careful proofreading of the manuscript and fixed. 

Author Response

Reviewer 5 comments

Author response

I read the manuscript with great interest. I liked the presentation style and discussion of the findings. A few things I may have to suggest. The authors must make sure that they proofread the revised version of the manuscript before uploading it to the journal website. I think almost all tables are incorrectly referred to in the main text, which could have been avoided, had the authors done a careful proofreading.

The manuscript has been proofread and sent to English Editor

Secondly, the authors also did not follow a single style when it came to referring to previous studies for in-text citations. For example, sometimes they just used the authors-year, as Moran et al (2022) in line 208. On other times, they used both, as "Lester et al, 2010 found that poor knowledge of guidelines resulted in an overall low implementation [29]" in lines 217-218.

References have been corrected using recommended reference style.

Also, I think, some sentences (e.g., "It is recommended that nurses receive training on UTT and its benefits to increase their knowledge and promote successful implementation in clinics." in lines 35-36) need to be rewritten or rephrased for clarity. For example, it was not clear to me from the way the sentence in lines 35-36 was constructed who was making the recommendation and/or on what basis. If the recommendation is based on the findings of the present study, then it should be made plain and clear. I am not going to list such sentences. However, I encourage the authors to look for sentences that have confusing construct and try to rewrite/rephrase them for clarity. Again, these things could be found in a careful proofreading of the manuscript and fixed.

Manuscript was sent to English Editor and proofread

Round 2

Reviewer 4 Report

Review- Round 2

1.  What is the reference for the "Universal-Test-and-Treat (UTT) guideline" that is used in your HIV clinics? The mentioned references are related to this topic but I didn't found a specific guideline or protocol of WHO. Please mark this reference.

2. How was constructed and validated the questionnaire? It is recommended by WHO? It should be included as supplementary material.

3. R167: "the" not "th" 
